# Considerations about the Continuous Assay Methods, Spectrophotometric and Spectrofluorometric, of the Monophenolase Activity of Tyrosinase

**DOI:** 10.3390/biom11091269

**Published:** 2021-08-25

**Authors:** Pablo García-Molina, José Luis Munoz-Munoz, Joaquin A. Ortuño, José Neptuno Rodríguez-López, Pedro Antonio García-Ruiz, Francisco García-Cánovas, Francisco García-Molina

**Affiliations:** 1GENZ-Group of Research on Enzymology, Department of Biochemistry and Molecular Biology-A, Regional Campus of International Excellence “Campus Mare Nostrum”, University of Murcia, Espinardo, 30100 Murcia, Spain; Pablo.garcia14@um.es (P.G.-M.); neptuno@um.es (J.N.R.-L.); canovasf@um.es (F.G.-C.); 2Microbial Enzymology Group, Department of Applied Sciences, University of Northumbria, Ellison Building A, Newcastle Upon Tyne NE1 8ST, UK; 3Department of Analytical Chemistry, Faculty of Chemistry, University of Murcia, 30100 Murcia, Spain; jortuno@um.es; 4Group of Chemistry of Carbohydrates, Industrial Polymers and Additives, Department of Organic Chemistry, Regional Campus of International Excellence “Campus Mare Nostrum”, University of Murcia, Espinardo, 30100 Murcia, Spain; pagr@um.es; 5Department of Anatomía Patológica, Hospital General Universitario Reina Sofía, Av. Intendente Jorge Palacios, 1, 30003 Murcia, Spain

**Keywords:** tyrosinase, polyphenol oxidase, monophenolase activity, fluorimetric method, spectrophotometric method, LOD^M^

## Abstract

With the purpose to obtain the more useful tyrosinase assay for the monophenolase activity of tyrosinase between the spectrofluorometric and spectrophotometric continuous assays, simulated assays were made by means of numerical integration of the equations that characterize the mechanism of monophenolase activity. These assays showed that the rate of disappearance of monophenol ($V_{\mathrm{ss}}^{M,M}$) is equal to the rate of accumulation of dopachrome ($V_{\mathrm{ss}}^{M,\mathrm{DC}}$) or to the rate of accumulation of its oxidized adduct, originated by the nucleophilic attack on *o*-quinone by a nucleophile such as 3-methyl-2-benzothiazolinone (MBTH), ($V_{\mathrm{ss}}^{M,\;A-\mathrm{ox}}$), despite the existence of coupled reactions. It is shown that the spectrophotometric methods that use MBTH are more useful, as they do not have the restrictions of the L-tyrosine disappearance measurement method, of working at pH = 8 and not having a linear response from 100 μM of L-tyrosine. It is possible to obtain low LOD^M^ (limit of detection of the monophenolase activity) values with spectrophotometric methods. The spectrofluorimetric methods had a lower LOD^M^ than spectrophotometric methods. In the case of 4-hydroxyphenil-propionic acid, the LOD^M^ obtained by us was 0.25 U/mL. Considering the relative sensitivities of 4-hydroxyanisole, compared with 4-hydroxyphenil-propionic acid, LOD^M^ values like those obtained by fluorescent methods would be expected.

## 1. Introduction

Tyrosinase (EC 1.14.18.1) is a cuproprotein that catalyses the hydroxylation of monophenols to *o*-diphenols (monophenolase activity) and the oxidation of *o*-diphenols to *o*-quinones (diphenolase activity), with the help of molecular oxygen [1]. This enzyme is distributed throughout the phylogenetic scale [2].

The enzyme tyrosinase, in the catalytic cycle, is found in three forms—meta, deoxy and oxy, which are differentiated by the degree of oxidation of copper: Em: metatyrosinase (Cu^2+^ Cu^2+^); Ed: deoxytyrosinase (Cu^1+^ Cu^1+^); Eox: oxytyrosinase (Cu^2+^Cu^2+^O_2_^−2^) [3]. In the case of mushroom tyrosinase, at the oxygen concentration existing in the solutions, the enzyme is saturated [4]. The two enzymatic forms Em and Eox are active on *o*-diphenols, oxidizing them to *o*-quinones, Em without oxygen and Eox with the participation of oxygen. However, with respect to monophenols, the Em form is inactive, forming an EmM dead-path complex. The Eox form is active on monophenols, carrying out its hydroxylation to *o*-diphenols through an electrophilic aromatic substitution [5,6,7]. This difference in activities of the Em and Eox forms is the origin of the kinetic complexity of the tyrosinase monophenolase activity. The EmM complex is inactive and therefore for the enzyme to be active on monophenols it is necessary for the Em form to be reduced to originate the Ed form and its binding to the oxygen to form Eox, closing the catalytic cycle (Figure 1) [5,6,7].

There is no such problem in diphenolase activity as the two forms Em and Eox are active on *o*-diphenol (D) (Figure 2) [5]. In nature, this problem has been solved (between the Em and Eox forms), as the product of the enzyme when it acts on its physiological substrates (L-tyrosine (M), L-dopa (D)) is *o*-Q that evolves to go through chemical reactions generating D in the medium, this D, L-dopa reduces Em to Ed and thus Eox (Figure 1) is formed [8].

When tyrosinase acts on M, it forms D, and this becomes *o*-Q, this through its chemical evolution originates dopachrome (DC) and accumulates D in the medium (Figure 1). That is, when the tyrosinase acts on M (monophenolase activity), the enzymatic reaction belonging to the diphenolase activity will occur but also the chemical reactions that involve evolution from *o*-Q [5].

This overlap between the activities (monophenolase and diphenolase) has led some authors to try to separate the monophenolase activity from the rest, in order to obtain the direct kinetic information of this enzymatic activity [9]. The first work that tried to separate the monophenolase activity used hydroxylamine to transform Em to Ed, and borate at pH = 8 to block the D originated by the enzyme from M. These authors make the experimental measurements of rates at the steady-state, measuring oxygen consumption (O_2_) [9].

Recently, based on the mechanism described above [9], it has been proposed as a measure of tyrosinase monophenolase activity to follow the disappearance of M directly [10,11,12]. According to these authors, thus, interference from the other processes (diphenolase activity and chemical reactions of evolution from *o*-dopaquinone) would be avoided. The methods proposed under different approaches are spectrofluorimetric [10,11,12] and, therefore, very sensitive. However, they have some limitations as detailed below.

Previously, reliable and sensitive continuous spectrophotometric methods have been developed [13,14]. The first contribution to establish a spectrophotometric measurement method for monophenolase activity, measuring the oxidation of the adducts formed between *o*-quinones and MBTH, based on the method described by Winder and Harris for diphenolase activity [13], was developed with mushroom tyrosinase and tyramine [14].

This method showed some advantages over those used up to that time:(a)Radiometric methods that are more sensitive but are discontinuous and require about 30 min per activity assay [13,15].(b)The spectrophotometric method normally used measuring DC [16]. This method is 8 times more sensitive and 6.5 times more sensitive than the proline-coupled assay [17].(c)Adducts and oxidized adducts caused by the coupling of MBTH with *o*-quinones show high molar absorptivity coefficients (see Appendix A).(d)The method has no interferences since the absorbance measurement occurs in the visible area, where neither the substrates nor the enzyme absorbed.(e)The adducts are stable at acidic pH and at pH > 5.5 they evolve presenting an isosbestic point, this particularity makes it possible to measure different enzymes of fruits and vegetables that have an optimal pH in the range between 4.0 and 5.0 [18].

Subsequently, this continuous spectrophotometric method was improved by demonstrating that p-hydroxyphenylpropionic acid was a better substrate for the enzyme and had a kcat 2.6 higher than tyramine [19]. Furthermore, because it is an acid, it does not show solubility problems.

Further optimization of the method was achieved using 4-hydroxyanisole as a substrate [18,20], making this substrate the best for a whole series of tyrosinases or polyphenol oxidases of fruits and vegetables such as apple, pear, strawberry, avocado, medlar and artichoke. The origin of the improvement of the continuous spectrophotometric method could be due to the higher catalytic constant that this substrate shows, corresponding to the lower value of chemical displacement of the C-4 carbon of this compound δ_4_ = 152.35 (p.p.m.). Subsequently, an extensive study on the substrate specificity of tyrosinase from mushroom, confirmed the results obtained with the rest of tyrosinases previously mentioned [21]. Note in Table 1 how the nature of the substrate affects the limit of detection (LOD^M^) values. Thus, the importance of the value of the chemical shift (δ_4_), obtained by C-4 ^13^C NMR, which in turn is related to the nucleophilia of the oxygen of the phenolic OH- group, is shown to predict the sensitivity of the method of measure.

The pH value used in the fluorometric methods was high (pH = 8) [10,11,12], to trap the *o*-diphenol generated in the reaction through the use of borate, thus it deviates from the optimum pH of the mushroom enzyme, pH = 6.8. For the same reason, spectrofluorimetric methods could not be applied to measure monophenolase activities of enzymes of fruits and vegetables that have their optimum pH between 4.0 and 5.0 (Apple (4.5), Avocado (4.6), pear (5), medlar (4.4) and strawberry (4.3)), the spectrophotometric method does not have any pH problem [18].

The sensitivity of the spectrofluorimetric method is shown in Table 1, with the low value of the LOD^M^ (0.0721 U/mL). With spectrophotometric methods, sensitivities in a range like spectrofluorimetrics could be achieved by varying the nature of the substrate (from L-tyrosine to 4-hydroxyanisole: see Table 1). Relationships are also obtained between the LOD values of different fruit and vegetable enzymes, depending on the type of substrate used, like those described in Table 1. These relationships ($R={\mathrm{LOD}}^{\mathrm{PHPPA}}/{\mathrm{LOD}}^{4\mathrm{HA}}$) are: Apple (3.78), artichoke (4.2), pear (3.91), medlar (3.60), avocado (3.76) and strawberry (3.82) [18].

Recently, related to the study on oculocutaneous albinism I, Isothermal titration calorimetry (ITC) has also been successfully used for the evaluation of kinetics of tyrosinase [22], and it is possible to extend the method in the presence of MBTH as shown for assaying manganese peroxidase [23].

The main purpose of this work is demonstrated, through simulated assays by numeric integration of the differential equations of the monophenolase activity of tyrosinase, that despite the coupled reactions that occur in the action of tyrosinase on L-tyrosine, it is true that $V_{\mathrm{SS}}^{M,M}=V_{\mathrm{SS}}^{M,\mathrm{DC}}$ and with respect to oxygen consumption it is true that $V_{\mathrm{SS}}^{M,O_{2}}=1.5V_{\mathrm{SS}}^{M,M}=1.5V_{\mathrm{SS}}^{M,\mathrm{DC}}=1.5V_{\mathrm{SS}}^{M,A-\mathrm{ox}}$, proving the usefulness of spectrophotometric versus fluorometric methods. Moreover, the fluorometric assays (10–12), are compared with the spectrophotometric published by our group [14,18,19,20,21,24,25].

## 2. Material and Methods

Experimental results of both, fluorometric [10,11,12], and spectrophotometric methods [14,18,19,20,21,24,25], shave been taken from the bibliography.

### 2.1. Materials

Tyrosine, dopa, tyramine hydrochloride, dopamine hydrochloride, 3,4-dihydroxyphenylpropionic acid (DHPPA), p-hydroxyphenylpropionic acid (PHPPA), 4-hydroxyanisol (HA) and 3-methyl-2-benzothiazolinone hydrazone (MBTH) were purchased from Sigma (Madrid). Stock solutions of the phenolic substrate were prepared in 0.15 mM phosphoric acid to prevent autooxidation. The acid characteristic of MBTH requires the use of 50 mM buffer in the assay medium [13].

### 2.2. Enzyme Source

Mushroom (Agaricus bisporus) tyrosinase (3900 units/mg) was purchased from Sigma and purified [26]. Protein content was determined by a modified Lowry method [27].

### 2.3. Spectrophotometric Assays

Absorption spectra were recorded in a Perkin-Elmer Lambda-2 spectrophotometer; online interfered with a PC laptop. Temperature was controlled at 25 °C using a Hacke DIG circulating water bath with a heater/cooler and checked with a precision of ±0.1 °C. We did not add substrate to the reference cuvettes with a final volume of 1 mL. To dissolve the MBTH-quinone adducts, 2% (by volume) N,N-dimethylformamide (DMF) was added to assay medium [14].

### 2.4. Kinetic Data Analysis

The methods used to monitor the monophenolase and diphenolase activities use MBTH to generate an adduct, which when is oxidized and generates a compound with high extinction coefficient (Appendix A). The precision of the method was checked by repeating the estimation de $V_{SS}^{M}$ (rate of action on monophenol) 10 times for three levels of ${\left[E\right]}_{0}$ 5.1 × 10^−4^, 2.6 × 10^−3^ and 5.2 × 10^−3^ unit/mL, the corresponding coefficients of variation being 5, 1.5 and 0.7%. The sensitivity of the method [28] was characterized by determining the detection limit (LOD^M^ = 3 × 10^−5^ unit/mL) and the quantification limit (LOQ^M^ = 7.2 × 10^−5^ unit/mL). In this work, the enzyme unit was defined as the amount that produced 1 µmol of the adduct/min when DHPPA was assayed [19]. The next step was the transformation of the monophenolase activity expressed in the activity units described above in the action of the enzyme on DHPPA, in the unit described in [10], which correspond to the amount of enzyme that consume 1 nmol of monophenol by minute or that produce 1 nmol of product by minute. Therefore, LOD^M^ = 3 × 10^−5^ unit/mL, which is equivalent for propionic p-hydroxyphenol to 3 × 10^−2^ U/mL, where U is the amount of enzyme that produces one nmol per minute. The relationships between catalytic constants [29] of the diphenol DHPAA and its monophenol PHPPA is 8.3, which means that one unit on DHPPA is equivalent to 8.3 of monophenolase units, so LOD^M^, expressed in monophenolase units, is LOD^M^ = 0.25 U/mL. Table 1 shows the relative sensitivities of different substrates (L-tyrosine, tyramine and 4-hydroxyanisole) with respect to our reference substrate (PHPPA), it is noteworthy that for 4-hydroxyanisole, values of LOD^M^ similar to those obtained with fluorescent methods, would be expected [18,20,21]. Thus, the sensitivity is related to the nucleophilic potency of the oxygen of the hydroxyl group attached at C-4, related to the value of the chemical shift (δ_4_) [21].

The initial rates were obtained by linear regression fitting portions of each experimental recording. These values were represented and fitted to the Michaelis–Menten equation using the Sigma Plot 9.0 program for Windows [30], providing the corresponding maximum rate (V_max_) and the Michaelis constant (K_M_) in each case.

### 2.5. Simulation Assays

The simulation through numerical integration of the differential equations that define the tyrosinase mechanism, helps to understand the kinetic behaviour of the enzyme and this help to compare different assay methods, both the fluorometric [10,11,12], and the spectrophotometric published by our group [14,18,19,20,21,24,25]. The simulated progress curves were obtained by numerical solution of the nonlinear set of differential equations corresponding to each kinetic scheme (see Appendix A), using arbitrary, but reasonable, sets of rate constants and initial concentration values and setting in the data entered, according to the different cases. The simulation assays showed the kinetic behaviour of the ligand and enzymatic species involved in the reaction mechanisms described here for tyrosinase. The numerical integration used a fourth order Range–Kutta method and the predictor corrector Adams–Moulton algorithm [31], implemented on a PC-compatible computer programme (WES) [32].

## 3. Results

The enzyme under study, tyrosinase, shows two activities: monophenolase and diphenolase.

### 3.1. Kinetic Analysis. Diphenolase Activity

The mechanism of diphenolase activity on D has been shown in (Figure 2), and in its structural form in Appendix A. In the steady state, it is shown by means of the deduction of the corresponding analytical expression. Rate of product formation ($V_{\mathrm{SS}}^{D,\mathrm{DC}}$) and substrate disappearance $V_{\mathrm{SS}}^{D,O_{2}}$, $V_{\mathrm{SS}}^{D,D}$ are the same (see Appendix A).

### 3.2. Kinetic Analysis. Monophenolase Activity

In this case, tyrosinase acts on M (enzymatic stage), generating D, this can be oxidized to *o*-Q (enzymatic stage), or it is released into the medium, subsequently, the *o*-Q evolves towards DC, regenerating D in the medium (enzymatic-enzymatic-chemical mechanism E_z_E_z_C) [16].

The detailed structural mechanism is shown in the Appendix A, in a simpler way it is described in Figure 1 and we can express it as in Figure 3.

This scheme shows the enzymatic stage of action of tyrosinase on M, its transformation into D, its oxidation to *o*-QH and the evolution of *o*-QH.

In the steady state, the quantity of matter entering the system of reactions will be shared between product and the sum the intermediates.

Considering that the rate of consumption of M in the steady-state is $V_{\mathrm{SS}}^{M,M}$ and that the rate of accumulation of DC in the steady state is $V_{\mathrm{SS}}^{M,\mathrm{DC}}$, it can be established if the system has reached steady state:(1)$$V_{\mathrm{SS}}^{M,M}=V_{\mathrm{SS}}^{M,\mathrm{DC}}$$

Moreover, the following material balance is fulfilled:(2)$$V_{\mathrm{SS}}^{M,M}t=V_{\mathrm{SS}}^{M,\mathrm{DC}}t=\left[\mathrm{DC}\right]+{\left[D\right]}_{\mathrm{SS}}+{\left[\mathrm{QH}\right]}_{\mathrm{SS}}$$

That is, the matter that enters the system is distributed among the different intermediaries and the final product. The intermediaries are D and *o*-QH, which reach steady-state with the levels ${\left[D\right]}_{\mathrm{SS}}$ and ${\left[\mathrm{QH}\right]}_{\mathrm{SS}}$. An analytical expression can be established for the rate of M consumption and for the accumulation of DC.

Applying the steady-state approximation for EoxD and EoxM in Figure 1, and taking into account the relationship of the velocities between the catalytic stages, as discussed above [8], it is fulfilled:(3)$$\left[\mathrm{DC}\right]=V_{\mathrm{SS}}^{M,\mathrm{DC}}\left(t-\frac{R{\left[M\right]}_{0}}{2k_{\mathrm{app}}V_{\mathrm{SS}}^{M,\mathrm{DC}}}-\frac{1}{2k_{\mathrm{app}}}\right)$$
with
(4)$$\frac{{\left[D\right]}_{\mathrm{SS}}}{{\left[M\right]}_{\mathrm{SS}}}=\frac{k_{5}k_{4}\left(k_{-6}+k_{7}\right)}{2k_{7}k_{6}\left(k_{-4}+k_{5}\right)}=R$$

DC accumulates in the medium according to the equation of a line Equation (3), the slope of this line is the steady-state rate $V_{\mathrm{SS}}^{M,\mathrm{DC}}=V_{\mathrm{SS}}^{M,M}$, the cut with the axis of time at a point t = τ, in which [DC] = 0, resulting in
(5)$$\tau =\frac{R{\left[M\right]}_{0}}{V_{\mathrm{SS}}^{M,\mathrm{DC}}}+\frac{1}{2k_{\mathrm{app}}}$$

After the lag period described by Equation (5) has finished, which is when the fulfilment of Equation (4) occurs, the rate has the analytical expression indicated by Equation (S9). Therefore, it is not necessary to measure substrate disappearance to obtain a correct measure of enzyme activity.

### 3.3. Simulation under Different Experimental Approximations of the Monophenolase Activity Mechanism

The systems of differential equations for monophenolase activity are shown in the Supporting Information. Although the objective of this work is the measurement of tyrosinase monophenolase activity, it is convenient to consider the two activities of the enzyme (for the diphenolase activity, see Appendix A).

#### Monophenolase Activity

The monophenolase activity of tyrosinase consists of the hydroxylation of monophenols to *o*-diphenols and the oxidation of these to *o*-quinones. There are several types of monophenols [24], and depending on the chemical evolution of the *o*-quinones generated, a certain kinetic behaviour of tyrosinase is obtained. In this work, we will focus on monophenols such as L-tyrosine, tyramine, 4-hydroxyphenylpropionic, 4-hydroxyanisole whose *o*-quinones either evolve generating D in the medium (L-tyrosine, tyramine) or they can be attacked by a nucleophilic reagent as MBTH and achieve the same objective of accumulation of D in the medium and in this way carry out the passage from Em to Eox, the first inactive on monophenols while the second is.

Through different simulation conditions, it is shown that in the steady state, the rate of consumption of M (L-tyrosine) ($V_{\mathrm{SS}}^{M,M}$) is equal to the rate of formation of DC ($V_{\mathrm{SS}}^{M,\;\mathrm{DC}}$), or in general a colored derivative that is generated by a nucleophilic attack by a reagent such as MBTH, followed by an oxidation reduction reaction, A-ox ($V_{\mathrm{SS}}^{M,A-\mathrm{ox}}$), in addition, $V^{M,O_{2}}$ is related to the other two rates by $V_{\mathrm{SS}}^{M,O_{2}}=1.5V_{\mathrm{SS}}^{M,M}=1.5V_{\mathrm{SS}}^{M,\mathrm{DC}}=1.5V_{\mathrm{SS}}^{M,A-\mathrm{ox}}$ in such way that the correct measurement of monophenolase activity can be done in theory by measuring the consumption of M or O_2_, or the formation of DC or A-ox. The simulation of the kinetic behaviour of the mechanism of monophenolase activity under different approaches helps us to confirm and establish with adequate experimental design.
(1)Action of tyrosinase on L-tyrosine. No *o*-diphenol (L-dopa) is added to the medium.The numerical integration of the set of differential equations that describe the mechanism (Figure 1) (see Appendix A), was carried out obtaining the following results.
(a)Accumulation of dopachrome, *o*-dopaquinone and *o*-diphenol in the medium. In Figure 1A, the DC, *o*-Q and D accumulation curves are shown. At short times, it is shown in Appendix A.(b)Rates of monophenol consumption ($V^{M,M}$), oxygen consumption ($V^{M,O_{2}}$) and dopachrome formation ($V^{M,\mathrm{DC}}$). In Figure 1B, the rates of consumption of M and O_2_ and the formation of DC are shown. Appendix A shows the curves of the velocities obtained in a short time.(c)Evolution over time of the different enzyme forms: Eox, Em and EmM. Appendix A shows the evolution of these enzymatic forms with time.(d)Variation of the concentration and rate of accumulation of *o*-diphenol with time. Appendix A shows the accumulation of D over time, until it reaches a steady-state with a constant level. Thus, Appendix A shows the accumulation of D at short times, continuously increasing towards the steady state. Appendix A shows the rate of D accumulation, it increases at the beginning of the reaction (step of Eox → Em), to decrease at longer times, until a value of the rate is equal to 0, in the steady state. At short times, in Appendix A you can see a burst in the rate of D accumulation, followed by a lag and then another burst to decrease going towards the steady state.(2)Action of tyrosinase on L-tyrosine adding enough *o*-diphenol to the medium to reach the steady-state.
(a)Accumulation of dopachrome, *o*-dopaquinone and *o*-diphenol in the medium. The curves shown in Figure 2A show the concentrations of *o*-Q, D and DC over time. In Appendix A the short-time curves are shown, a small lag can be seen in the accumulation of DC and a burst in the accumulation of *o*-Q.(b)Rates of consumption of monophenol and oxygen and rate of dopachrome formation. In Figure 2B the consumption rates of O_2_, M and the rate of DC formation are shown and in Appendix A, we show the same parameters but at short times.(c)Evolution over time of the different enzyme species. Appendix A shows the evolution of the different species enzymes.(d)Variation of the concentration and rate of accumulation of *o*-diphenol with time. It is shown in the Appendix A.(3)Action of tyrosinase on L-tyrosine, adding before starting the reaction an amount of *o*-diphenol, less than that necessary to reach steady state.
(a)Accumulation of dopachrome, *o*-dopaquinone and *o*-diphenol in the reaction medium. Appendix A show the accumulation of DC, *o*-Q and D. In Appendix A, you can see the detail in short time.(b)Rates of consumption of monophenol and oxygen and rate of dopachrome formation. Rates are shown in Appendix A and at short times in Appendix A.(c)Evolution over time of the different enzyme species. A graph like that of Appendix A is obtained. The enzyme accumulates mostly as EmM.(d)Variation of *o*-diphenol concentration with time. Appendix A shows D variation.(4)Action of tyrosinase on L-tyrosine, adding before starting the reaction an amount of *o*-diphenol greater than that necessary to reach steady state. The speeds at long and short times are shown in Appendix A, respectively.(5)Action of tyrosinase on L-tyrosine in the presence of a nucleophile N.
(a)The curves obtained in this section are like those obtained in Section 1, the only difference is that the *o*-Q evolution constant becomes much larger as the nucleophilic attack of N is more powerful than the cyclization of *o*-Q. In the action of tyrosinase on M in the presence of N, Figure 3A shows the accumulation of *o*-Q, D and A-ox with time. It shows a burst for *o*-Q and D and a very small lag for A-ox.(b)Rates of consumption of M, O_2_ and the formation of A-ox. In Figure 3B, the curves obtained for the rate of consumption of M and O_2_ and that of accumulation of A-ox are shown.(c)Enzyme species evolve in a similar way to the other cases and the enzyme accumulates fundamentally as EmM, Figure 3C.(d)Variation of the concentration and rate of accumulation of D over time. It is shown in Figure 3D.

## 4. Discussion

The analysis of the mechanism of monophenolase activity of tyrosinase under different initial conditions was carried out by simulation by numerical integration of each particular case.

### 4.1. Action of Tyrosinase on L-Tyrosine. No o-Diphenol (L-Dopa) Is Added to the Medium

Figure 1A shows that DC accumulates with a lag (τ), while the *o*-Q and D accumulate with a burst. When the system reaches the steady state ($t_{99}≅\;4.6\;\tau$), the slope of the DC accumulation line corresponds to the steady-state rate ($V_{\mathrm{SS}}^{M,\mathrm{DC}}$). Appendix A shows the accumulation of these species at short times increasing to reach the steady state.

Rates of consumption of O_2_ and M are higher at the beginning of the reaction (Figure 1B), because in the native enzyme there is approximately 30% in the Eox form and this species consumes M and O_2_. The rate of DC accumulation increases with time, in the steady state:$V_{\mathrm{SS}}^{M,M}=V_{\mathrm{SS}}^{M,\mathrm{DC}}$. However, the rate of O_2_ consumption ($V_{\mathrm{SS}}^{M,O_{2}}$) in the steady-state is 1.5 times $V_{\mathrm{SS}}^{M,M}$ or $V_{\mathrm{SS}}^{M,\mathrm{DC}}$. The explanation for these curves is that at steady-state the enzyme molecule performs two cycles in the hydroxylase pathway (see Figure 1) consuming 2M and 2O_2_ and one cycle in the oxidase pathway consuming 2D and 1O_2_. The two molecules of M originate 2Q and those of 2D other 2Q, which regenerate between 4Q, 2 of D and 2 of DC, for this reason, as shown in Figure 1B ($V_{\mathrm{SS}}^{M,O_{2}}=1.5V_{\mathrm{SS}}^{M,M}=1.5V_{\mathrm{SS}}^{M,\mathrm{DC}}$) [8,33]. Note a less abrupt drop in the rate of consumption of M and O_2_ and an increase in the rate of formation of DC (Appendix A), because at t = 0 there is approximately a 30% enzyme in Eox form that reacts rapidly by consuming M and O_2_ and practically all the enzyme is found as EmM, that is, inactive, as will be seen later. The appearance of lag in the rate of DC formation can be observed.

The assays shown in Appendix A are explained at the beginning of the reaction and the enzyme is mostly found as: 70% Em and 30% Eox, when adding M, the Eox form reacts rapidly generating *o*-Q and passing to Em (inactive on M) (Appendix A). The Em form reacts with M to form the inactive complex EmM, the simulated curves show how the enzyme is mostly as EmM.

When tyrosinase begins its action on M, a lag period, τ, is originated in the accumulation of DC and at the same time the system accumulates D in the medium, the lag period is related to the time in which the system reaches the stationary phase and that assumes a constant value of D. D variation with time is shown in Appendix A.

### 4.2. Action of Tyrosinase on L-Tyrosine Adding Enough o-Diphenol to the Medium to Reach the Steady State

This situation assumes that the system does not have to accumulate D in the medium and thus the steady state is reached earlier at t → 0. D quantity is given by Appendix A.

Note that now the DC accumulates without a delay period, the steady state is reached at t → 0 (Figure 2A and Appendix A). The steady state is reached quickly, M and O_2_ arrive before DC, due to the evolution of the *o*-Q. In the steady state, $V_{\mathrm{SS}}^{M,M}=V_{\mathrm{SS}}^{M,\mathrm{DC}}$ and $V_{\mathrm{SS}}^{M,O_{2}}=1.5V_{\mathrm{SS}}^{M,M}=1.5V_{\mathrm{SS}}^{M,\mathrm{DC}}$ (Figure 2B). A similar effect is seen in short-time records (Appendix A). The concentration of the enzymatic species EmM, Em and Eox shows that most of the enzyme is in the complex of dead-path EmM (Appendix A). The variation of D with time is very small because practically the system is in the steady-state (Appendix A).

### 4.3. Action of Tyrosinase on L-Tyrosine, Adding before Starting the Reaction an Amount of o-Diphenol, Less Than That Necessary to Reach the Steady State

The system will accumulate the D resulting from the difference between the D of the steady state and the added at the beginning. Thus, the lag in DC accumulation will be less than the described in Figure 2A (Section 4.1) and the steady-state will be reached earlier.

DC is accumulated with a lag that is lower than in the absence of added D and the *o*-Q and D accumulate with a burst, until reaching the steady state (Appendix A). The burst of the *o*-Q moves in time because it must be generated and evolve towards DC (Appendix A). In long time, the system reaches the steady state and then it is satisfied that $V_{\mathrm{SS}}^{M,M}=V_{\mathrm{SS}}^{M,\mathrm{DC}}$ and with respect to O_2_: $V_{\mathrm{SS}}^{M,O_{2}}=1.5V_{\mathrm{SS}}^{M,M}=1.5V_{\mathrm{SS}}^{M,\mathrm{DC}}$ (Appendix A). At short times, no rate has reached the steady state and appreciable differences are seen between $V^{M,M}$ and $V^{M,\mathrm{DC}}$ (Appendix A). D concentration reaches the steady state before than the described in Appendix A (Section 4.1).

These conditions should be of choice to work experimentally with monophenolase activity. In order to reduce lag, a small amount of *o*-diphenol should be added before starting the reaction. When the substrate concentration is varied, the *o*-diphenol concentration must be varied such that the ratio is constant.

### 4.4. Action of Tyrosinase on L-Tyrosine, Adding before Starting the Reaction an Amount of o-Diphenol Greater Than That Necessary to Reach Steady-State

The system advances towards the steady state decreasing the concentration of D until reaching that of the steady-state, and then fulfilling the relationships between the rate: $V_{\mathrm{SS}}^{M,M}=V_{\mathrm{SS}}^{M,\mathrm{DC}}$ and $V_{\mathrm{SS}}^{M,O_{2}}=1.5V_{\mathrm{SS}}^{M,M}=1.5V_{\mathrm{SS}}^{M,\mathrm{DC}}\;$(Appendix A), and at short times (Appendix A). The rest of the parameters follow a behavior like the described in Section 4.1, Section 4.2 and Section 4.3.

### 4.5. Action of Tyrosinase on L-Tyrosine in the Presence of a Nucleophile N

The measurement of the activity of the enzyme on M in the presence of MBTH has led to the design of very sensitive measurement methods of the monophenolase and diphenolase activities, and this has allowed to characterize especially the monophenolase activity in fruits and vegetables that was described that many polyphenol oxidases had no monophenolase activity [13,14,25].

In this case, the arrival to the steady state is advanced and again it is true that the rate of formation of the colored adduct, the oxidized form of the adduct (A-ox) is related to the rate of consumption of M and O_2_ by the relationship $V_{\mathrm{SS}}^{M,M}=V_{\mathrm{SS}}^{M,\;A-\mathrm{ox}}$ and $V_{\mathrm{SS}}^{M,O_{2}}=1.5V_{\mathrm{SS}}^{M,M}=1.5V_{\mathrm{SS}}^{M,A-\mathrm{ox}}$ (Figure 3A,B). Evolution of D and the enzymatic species is shown in Figure 3C,D. Figure 3D shows the variation in the rate of accumulation of D over time. Initially there is a burst in the rate of formation of D, and then at long times decrease the rate until it vanishes in the steady state.

In all the cases described, it is shown that the measurement of monophenolase activity in tyrosinase acting on M can be done either by measuring the disappearance of M or O_2_, or by measuring the formation of DC or A-ox, when the system is in steady-state.

### 4.6. Comparison of the Measurement of Monophenolase Activity of Tyrosinase Obtained by Continuous Spectrophotometric Methods That Use MBTH as a Nucleophilic Reagent with Continuous Fluorometric Methods That Measure the Disappearance of L-Tyrosine through Quenching L-Dopa Fluorescence by Borate

Fluorometric methods are normally more sensitive than spectrophotometric ones, but when it comes to measuring the activity of an enzyme, in this case tyrosinase, one must consider several variables such as pH, substrate concentration and measurement interferences. Proposed continuous spectrophotometric methods to measure tyrosinase monophenolase and diphenolase activity may have some advantages over spectrofluorimetric:
In spectrophotometric methods, product formation is measured in an area where substrates do not interfere and it is more sensitive to measure product formation than substrate disappearance [24]. In the proposed spectrofluorimetric methods, the disappearance of substrate L-tyrosine, is measured.In these spectrophotometric methods, product formation is measured. Moreover, there are no phototube saturation problems. Thus, the substrate concentration can be increased because it does not interfere. In the spectrofluorimetric methods proposed, when measuring the disappearance of substrate (L-tyrosine), by increasing its concentration, linearity can be lost, whereas in reality, it occurs in such a way that in the proposed methods [10,11,12], which follow fluorescence, only the behavior is linear up to a concentration of 100 µM of L-tyrosine.What is indicated in Section 2 is very important when characterizing the kinetics of tyrosinase monophenolase activity, as the concentration cannot be increased more than 100 µM. Indeed, if the initial velocity values are adjusted, with respect to the L-tyrosine concentration, erroneous kinetic parameters are obtained, because a true hyperbolic dependence of the $V_{\mathrm{SS}}^{M,M}\;\mathrm{vs}.\;{\left[M\right]}_{0}$ values has not been obtained. Thus, the authors working with mushroom tyrosinase, obtain a value of K_M_ = 19.51 µM [10,11,12], when the value found for this enzyme is between 0.21 mM [21] and 0.31 mM [34]. At high pH values, K_M_ values are constants [35]. In general, this problem is encountered by most of the methods that attempt to measure the disappearance of substrate [24].The lack of linearity between the fluorescence measurement and the concentration of L-tyrosine above 100 µM make kinetic studies difficult and especially in the presence of inhibitors. Thus, when zinc ion is studied as a tyrosinase inhibitor [10], data are obtained that indicate that it is a competitive inhibitor, so in order to determine the K_I_ value, the concentration of L-tyrosine must be increased by more than 100 µM, like this not possible, abnormal results are obtained. On the other hand, the IC50 value described is 14.64 µM and the K_I_ value = 67.60 µM [10], but in the case of a competitive inhibitor, it can be shown that IC50 > K_I_, specifically IC50 = (n+1)K_I_, where n is the ratio of substrate concentration to Km [36]. This relationship between IC50 and K_I_ is well established in the data shown in Tables 1 and 2 in [37] and in Table 1 in [38].The relationship of rates of consumption of M, O_2_ and formation of DC described in Table S5 of the Supporting Information [10] shows in the monophenolase activity the equality of rates of consumption of O_2_ and consumption of M, however the DC formation rate is not comparative; this aspect confirms the validity of the measure measuring O_2_ consumption that was described previously [9]. The stoichiometry of diphenolase activity predicts that the O_2_ consumption rate should be equal to the formation rate of DC, as demonstrated previously [8]. In the first column (substrate + TYR, diphenolase) of Table S5 [10], the described relationship is not fulfilled; however, it is achieved in the last column of Table S5 [10] (Substrate + TYR + hydroxylamine).A feature of spectrofluorimetric methods that measure the disappearance of M could be the one that reaches the steady-state earlier, as D does not accumulate in the medium under these test conditions. This aspect can be corrected in spectrophotometric methods by adding a small concentration of D to the medium (${\left[D\right]}_{\mathrm{added}}<{\left[D\right]}_{\mathrm{SS}}$) (Figure 2A) to decrease the lag period. Furthermore, it is thus achieved that the substrate concentration varies very little and therefore measured initial rates will correspond to the steady-state rate.

## 5. Conclusions

The numerical integrations carried out in this work revealed that to correctly measure the monophenolase activity of tyrosinase, we can measure the disappearance of substrate M (fluorimetrically) or O_2_ (oximetrically) in a medium at pH = 8 with borate, or either by spectrophotometrically measuring the appearance of DC or A-ox product, always considering that the system is in a steady state. Thus, if the system is in a steady state, the rate of substrate disappearance is equal to the rate of product formation, if they are initial steady-state rates. In this sense, the proposed spectrophotometric methods measuring the formation of oxidized adducts between *o*-quinones and MBTH, can be considered more suitable to follow the monophenolase activity of tyrosinase, taking into account the following aspects: (a) There are no interferences from substrates. (b) There is no interference from the enzyme. (c) The substrate concentration can be varied. (d) It is possible to work at different pH values and thus the enzymes of fruits and vegetables can be characterized. (e) The addition of a small amount of *o*-diphenol is convenient to decrease the delay period (τ), but not to suppress it.

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
