# Peer review of "Considerations about the Continuous Assay Methods, Spectrophotometric and Spectrofluorometric, of the Monophenolase Activity of Tyrosinase"

_biomolecules, 2021, doi:10.3390/biom11091269_

Round 1

Reviewer 1 Report

We are satisfied with the rebuttal and the revised version submitted by the authors.

However, we are not agree on one issue regarding the application of ITC for tyrosinase assay. We agree that the mechanism of tyrosinase and MnP are different so that similar analysis involving numerical integration to make simulated assays for MnP are not feasible if someone try to replicate them. However we did not propose a comparison of peroxidase and tyrosinase activities and/or mechanisms or new experiments, but simply said that it is suitable to mention ITC in the Introduction section along with other assay methods. ITC is very suitable for carrying out tyrosinase assays not only for carrying out reliable Michaelis-Menten kinetics but also for avoiding some obstacles indicated in the text. A paper has recently been published (Young et al. 2020). ITC exploit heat signal and is independent of the enzyme mechanism for assaying the enzyme. Authors have listed various published methods in the text that have been used to assay tyrosinases. We suggest adding in the Introduction section, a sentence about the use of ITC so that the readers are aware of this alternative.

Kindly use the following sentence as a guide:

“Isothermal titration caorimetry (ITC) has also been successfully used for the evaluation of kinetics of tyrosinase (Young et al. 2020) and it is possible to extend the method in the presence of MBTH as shown for assaying manganese peroxidase (Ertan et al. 2012).”

Young KL II, Kassouf C, Dolinska MB, Anderson DE, Sergeev YV. Human Tyrosinase: Temperature-Dependent Kinetics of Oxidase Activity. Int J Mol Sci. 2020;21(3):895.doi:10.3390/ijms21030895

Ertan H, Siddiqui KS, Muenchhoff J, Charlton T, Cavicchioli R. Kinetic and thermodynamic characterization of the functional properties of a hybrid versatile peroxidase using isothermal titration calorimetry: Insight into manganese peroxidase activation and lignin peroxidase inhibition. Biochimie. 2012 May;94(5):1221-31. doi: 10.1016/j.biochi.2012.02.012. PMID: 22586704.

Minor editing:

By the way, the main text contains a sentence written in Spanish, possibly remained from the authors' revision.

Author Response

We would like to thank the Reviewers for the constructive comments, and we have answered all suggestions point by point:

  • As stated by the Reviewer, we have incorporated the paragraph suggested by the Reviewer in line 144. In addition, we have incorporated both references suggested the Reviewer too.
  • We have translated the remaining sentences in Spanish that, by mistake, we submitted in the previous version.

Reviewer 2 Report

Glad to see the paper has been improved.

Author Response

We would like to thank the Reviewers for the constructive comments.

This manuscript is a resubmission of an earlier submission. The following is a list of the peer review reports and author responses from that submission.

Round 1

Reviewer 1 Report

The manuscript describes considerations about the measure for the monophenolase activity of tyrosinase. However, the manuscript is not written in a readable way. The reviewer is sorry to say it’s very hard to follow the authors. There many sentences are unclear, for example, line 20, 29, 32, 56, 77, 84, 122, 413, 496, 500. The reviewer fails to receive the messages the authors want to convey. It’s unclear why the research is important and what are the novelties. The reviewer doesn’t feel the manuscript could be revised in a short time to be suitable for publication.

Minor issues:

  • Scheme 1 is inappropriate, as the elements between the arrows are not balanced.
  • It would be nice if some experimental details from the SI are relocated to the main text.
  • Sources of the enzyme in Table 1 should be provided.

Reviewer 2 Report

Reviewers’ recommendations for authors:

Major points:

1) The title should have been more informative in line with the aim of the study.

2) The abstract should clearly summarize the whole study including principal methods employed (i.e., experimental, simulation assays etc.), aims (i.e., to determine which is the more useful tyrosinase assay), and the gains and main conclusions of this study. The hypothesis tested in this study should also be clearly stated.

3) It should be pointed out upfront that it’s a pure theoretical study (i.e., simulation assays) based on previously generated experimental data. Reading M&M sections in the main and supplementary documents give the impression that the experimental data has been generated in this study.

4) The text near the end in the R&D Section give the background information as to WHY the study was undertaken. This information (between the lines 380-417 and 464-478) should be in the Introduction section where the pros and cons of the existing published assays should be given upfront.

5) At the end of Introduction section, it should be mentioned as HOW the main hypothesis was tested in this study.

6) M&M sections should state exactly HOW this study was done? If previous experimental data were used, it should be mentioned along with the references.

7) R&D section can be reorganized so that it is easier to follow. We strongly suggest separating Results and Discussion sections so that it is easier to follow. Results section should only have data relevant to the current study and previously published results should not include.

8) Emphasis should be given to the assay (such as MBTH) which has been proposed as the best. Other peripheral results should be kept to minimum as necessary to understand the rationale and the flow of the study.

9) In Discussion section, authors should compare their own results with the relevant previously published articles. However, when we looked at the R&D section in the manuscript, we are unable to find this sort of discussion but confusing information. As the main objective of the study is to measure the monophenolase activity of the enzyme, the excessive information regarding diphenolase activity may be considered to reduce the impact of the study. This information can be limited to the introduction and supplementary material. In conclusion, there is a lot of unnecessary and unclear information in R&D section that stifles the novelty and conclusions of the study.

10) One important issue that should be considered in the revised manuscript is the inclusion of calorimetric assay in the discussion in R&D, Conclusion or Future avenues sections. Because manganese peroxidase assay also requires MBTH, kindly comment in the revised paper if similar rationale is valid in case tyrosinase is assayed using isothermal titration calorimetry (ITC)? (See Ertan, H. et. al., 2012 Biochimie. 94(5):1221-1231.)

11) Authors were referring too much to the Supp. Material in the main text. As a rule, all essential information (e.g., At least basic protocols for the enzyme assays) should be in the main text and only large data and secondary information should be included in Supp. material.

12) As most of the methods for tyrosinase assays discussed in the present study are similar to what authors already published before (Garcia-Molina et al. 2007 J. Agric. Food Chem. 55:9739–9749), the authors must state in the revised paper and covering letter about the NOVELTY of this study. Only Novel aspects be included in the revised manuscript with novel features highlighted in the abstract.

Minor points:

1) Line 30: Delete "those" and insert: "the spectrometric assays had lower LODM."

2) Line 56: “it is” should be “its”.

3) Line 57: “binding the oxygen to pass to Eox” should be “binding to the oxygen to form Eox”.

4) Line 63: “to through” should be “to go through”.

5) Rewrite the sentence in lines 69-70: “That is, when the enzyme…”.

6) Rewrite the phrase in line 85: “very sensible…” (e.g., Sensible or sensitive?)

7) Line 94: Should be “spectrofluorimetric” instead of “spectrophotofluorimetric”.

8) Line 113: Should be “…results were transferred…”

9) In equation 2: Make sure of all subscripts (e.g., kapp)

10) Line 220: Delete “tried”.”

11) Rewrite the phrase in line 77: “…originates from the enzyme from M.”

12) Line 84-85: “above” should be “below”. It would be better to rewrite the whole sentence.

13) Line 140-141. Rewrite the sentence starting with “Thus…”. Incomplete sentence. Should be rewritten.